# UV-C Promotes the Accumulation of Flavane-3-ols in Juvenile Fruit of Grape through Positive Regulating *VvMYBPA1*

**DOI:** 10.3390/plants12081691

**Published:** 2023-04-18

**Authors:** Jinjun Liang, Jianyong Guo, Yafei Liu, Zening Zhang, Runtian Zhou, Pengfei Zhang, Changmei Liang, Pengfei Wen

**Affiliations:** 1College of Horticulture, Shanxi Agricultural University, Taigu 030801, China; liangjinjun1989@163.com (J.L.);; 2College of Information Science and Engineering, Shanxi Agricultural University, Taigu 030801, China

**Keywords:** grape, UV-C, *VvMYBPA1*, MBW complex, flavonoids

## Abstract

Flavane-3-ol monomers are the precursors of proanthocyanidins (PAs), which play a crucial role in grape resistance. Previous studies showed that UV-C positively regulated leucoanthocyanidin reductase (LAR) enzyme activity to promote the accumulation of total flavane-3-ols in juvenile grape fruit, but its molecular mechanism was still unclear. In this paper, we found that the contents of flavane-3-ol monomers increased dramatically at the early development stage grape fruit after UV-C treatment, and the expression of its related transcription factor *VvMYBPA1* was also enhanced significantly. The contents of (−)-epicatechin and (+)-catechin, the expression level of *VvLAR1* and *VvANR*, and the activities of LAR and anthocyanidin reductase (ANR) were improved significantly in the *VvMYBPA1* overexpressed grape leaves compared to the empty vector. Both VvMYBPA1 and VvMYC2 could interact with VvWDR1 using bimolecular fluorescence complementation (BiFC) and yeast two hybrid (Y2H). Finally, *VvMYBPA1* was proven to bind with the promoters of *VvLAR1* and *VvANR* by yeast one hybrid (Y1H). To sum up, we found that the expression of *VvMYBPA1* increased in the young stage of grape fruit after UV-C treatment. VvMYBPA1 formed a trimer complex with VvMYC2 and VvWDR1 to regulate the expression of *VvLAR1* and *VvANR*, thus positively promoting the activities of LAR and ANR enzyme, and eventually improved the accumulation of flavane-3-ols in grape fruit.

## 1. Introduction

Proanthocyanidins (PAs) are important secondary metabolites of grape [1]. Flavan-3-ol monomers, as the precursor of PAs, play a key role in the biosynthetic pathway of PAs. However, it could be decomposed to produce anthocyanidins, which was also crucial to the quality of grape berries and processed wine [2]. PAs played a crucial role in plant diseases resistance [3], such as reducing the effects of free radicals, ultraviolet light and oxidation [4,5]. In addition to differences in grape varieties, the synthesis of grape PAs was also affected by many external environments, such as ultraviolet light. Therefore, it is of great significance to explore the molecular mechanism of PAs biosynthesis in UV-C treated grape fruits.

UV-C refers to the ultraviolet ray with a wavelength in the range of 200–280 nm, also known as short wave sterilizing ultraviolet ray. Its penetration ability is very weak, and it can hardly penetrate most transparent glass and plastic products. There were two reasons to enhance plant stress resistance after UV-C irradiation treatment. On the one hand, it could change the ultrastructure of plant cells [6]; on the other hand, it could induce the natural propane metabolic pathway in the body to produce phenols [7]. Studies had shown that UV-C radiation could cause the expression of stilbene synthase gene *VvSTS15/21*, which led to the accumulation of stilbene compounds in grapes and simulated the infection of pathogenic bacteria to a certain extent to improve disease resistance [8]. Niu et al. indicated that UV-C inhibited the growth and development of grape fruit, thereby delaying the turning stage [9]. The significantly differential transcription factors *VvMYBPA1*, *VvMYB4*, *WRKY57-like*, *VvMYB14*, and *WRKY65* were screened out through transcriptome analysis of grape blades before and after UV-C radiation treatment [10].

Leucoanthocyanidin reductase (LAR) and anthocyanidin reductase (ANR), which were two unique key enzymes in the PAs synthesis pathway, generated the (+)-catechin and (−)-epicatechin, respectively [11]. R2R3-MYB transcription factors played a key role in regulating biosynthesis of PAs in horticultural plant, such as *MdMYB9/11, PpMYB134*, *CsMYB60*, and *DkMYB19/20*, etc. [12,13,14,15]. The biosynthesis of PAs was regulated by these transcription factors through regulating the expression of key enzymes in the pathway, especially ANR and LAR. Ectopic overexpression of *VvMYBPA1* compensated for the phenotype of mutant *tt2* in *Arabidopsis* [16], but no homologous gene was found in *Arabidopsis*.

Previous studies had indicated that R2R3-MYB usually combined with WD40-repeat (WDR) and basic helix loop helix (bHLH) to form a MBW trimer complex, and then regulated the biosynthesis of flavonoids and anthocyanin [17]. In *Arabidopsis*, AtTT2 (MYB) interacted with AtTTG1 (WD40) and AtTT8 (bHLH) to generate a MBW complex and improved the content of PAs by directly promoting the expression of *AtBANYULS* (*AtBAN*), which is the PAs synthesis structural gene in *Arabidopsis* seed coat [18]. In cucumber, CsMYB60 formed a complex with CsbHLH42/MYC1 and CsWD40 to regulate the synthesis of PAs [14]. DkMYB4, DkMYC1, and DkWDR1 formed a MBW complex in persimmon, and finally regulated the accumulation of PAs by affecting the expression of *DkANR* gene in persimmon fruit [19]. FaMYB9/FaMYB11 could regulate the expression of *FaANR* and *FaLAR* through forming a trimer complex with FaTTG1 and FabHLH3, and finally regulated the accumulation of PAs in strawberry [20]. The complex formed by VvMYBA1, WDR1, and MYC1 regulated the synthesis of Yan73 grape anthocyanin in grape [21]. In addition, *VvWRKY26* might participate in a MBW trimeric complex to advance the accumulation of anthocyanins in grape [22]. These studies indicated that MBW complexes participated in regulating the biosynthesis of anthocyanidins and PAs in various horticultural plants.

Previous studies had showed that UV-C irradiation positively regulated VvLAR enzyme activity to promote the accumulation of total flavane-3-ols in early grape fruits [23], but its molecular regulation mechanism was still unclear. At present, studies on the regulation of grape PAs accumulation by a MBW trimer complex have been reported [24,25]. In this study, the different period grape fruits after UV-C irradiation treatment were used as experimental materials to explore the molecule mechanism of flavane-3-ol monomers in grapes. The biosynthesis-related transcription factor *VvMYBPA1* was further studied to elucidate its regulatory mechanism for the biosynthesis of flavane-3-ols.

## 2. Results

### 2.1. UV-C Led to the Accumulation of Flavane-3-ol Monomers in Juvenile Fruit of Grape

The previous research found that UV-C radiation improved content of VvLAR1 protein and LAR1 enzyme activity and eventually led to an increase in total flavane-3-ols content in the young grape fruit [23]. In this paper, the results indicated that the contents of (−)-epicatechin and (+)-catechin after 5 min UV-C radiation treatment were visibly higher than that in the control group at the early development stage of grape fruit (within 50 days after flowering). The contents of (−)-epicatechin of grape fruits, which were treated with UV-C radiation for 5 min at 20, 30, 40, and 50 days after flowering, were 43.32, 25.33, 18.23, and 16.18 µg·g^−1^, respectively, while those of the control group were 31.56, 18.56, 14.32, and 11.56 µg·g^−1^, respectively (Figure 1A). The contents of (+)-catechin of UV-C radiation treated grape fruits were 115.32, 65.33, 58.23, and 37.98 µg·g^−1^, respectively, at 20, 30, 40, and 50 days after flowering, while those in the control group were 80.56, 45.56, 37.32, and 27.56 µg·g^−1^, respectively (Figure 1B). The contents of (−)-epicatechin and (+)-catechin were significantly increased by UV-C radiation treatment in the young grape fruit. The flavane-3-ol monomers decreased significantly and reached the lowest after 70 days of flowering and remained basically unchanged. In a word, UV-C radiation promoted the contents of flavane-3-ol monomers at the young grape fruit stage.

We found that the expression of *VvMYBPA1*, which was treated by UV-C radiation for 5 min on the 20th day after flowering, was 3.12 times that of the control group and 1.25 times on the 30th day after flowering. After that, the expression of *VvMYBPA1* began to decrease, and UV-C radiation reduced its expression contrast to the control group (Figure 1C, Appendix A). In conclusion, the expression of *VvMYBPA1* had a high correlation with the contents of (−)-epicatechin and (+)-catechin after UV-C treatment at different stages of fruit development.

### 2.2. Characteristic Analysis of Transcription Factor VvMYBPA1

The phylogenetic analysis of VvMYBPA1 amino acids and related amino acids of other species was carried out by using the Neighbor-Joining method. It was found that VvMYBPA1 had the strongest similarity with MrMYB5 and DkMYB4, while VvMYBPA2 had the strongest similarity with AtTT2 (Figure 2A). We took the fusion vector 35S::*VvMYBPA1*-GFP as the experimental group, 35S:: GFP as the control group, and carried out subcellular localization in tobacco leaves. It was found that the green fluorescent signals in the experimental group were located in the nucleus, but the green fluorescent signals were dispersed in the whole cells in the control group (Figure 2B). The results showed that VvMYBPA1 protein was located in the nucleus and had weak similarity with AtTT2.

### 2.3. Instantaneous Overexpression of VvMYBPA1 Led to the Accumulation of Total Flavane-3-ols in Grape Leaves

In order to find out the reason as to why *VvMYBPA1* led to the content of total flavane-3-ols in grape, we conducted an experiment on instantaneous overexpression of *VvMYBPA1* in grape leaves. The levels of flavane-3-ol monomers, the activities of related enzymes, and the expression levels of related genes were measured using grape leaves with transient overexpression of *VvMYBPA1* and empty vector. The results showed that the grape leaves with transient overexpression of *VvMYBPA1* gene were darker than the control group after DAMAS staining (Figure 3A), which proved that the content of PAs increased after overexpression of the *VvMYBPA1* gene.

The amounts of flavan-3-ol monomers in the transient overexpression of *VvMYBPA1* group and the control group was determined by HPLC. The results showed that (+)-catechin increased from 36.33 in the control group to 71.83 µg·g^−1^ after overexpression of *VvMYBPA1*; The contents of (−)-epicatechin increased to 5.33 µg·g^−1^ from 2.59 in the control group (Figure 3B). The relative expression of related genes was measured by fluorescence quantification. It was found that compared to the empty vector overexpression leaves, the expression levels of PAs biosynthesis-related genes *VvANR*, *VvMYBPA1*, *VvLAR1*, and *VvLAR2* were significantly improved in the grape leaves with transient overexpression of *VvMYBPA1* (Figure 3C, Appendix A). We also found that the enzyme activities of LAR and ANR of the overexpressing *VvMYBPA1* grape leaves were increased significantly (Figure 3D,E).

### 2.4. VvMYBPA1 Formed a Trimer Complex with MYC2 through WDR1

Further exploring the molecular mechanism of *VvMYBPA1* promoted the content of total flavane-3-ols in grape. The interactive relationship between the VvMYBPA1 protein, the WDR1 (WD40 protein), and the MYC2 (bHLH protein) were further analyzed by Y2H and BiFC experiments in this paper.

Y2H results indicated that the positive group and the co-transformed colonies of VvMYBPA1/VvWDR1, VvMYBPA1/VvMYC2, and VvWDR1/VvMYC2 could grow on the four deficient medium (SD/- Trp-Leu-His-Ade) plate. The positive group and three test groups turned blue when grown on SD/- Trp-Leu-His-Ade/X-α-gal plate, which indicated that the VvMYBPA1 protein could interact with VvWDR1, VvMYC2 protein, and VvWDR1 could interact with MYC2 protein in vitro (Figure 4A). The results found that yellow fluorescence signals appeared in the nucleus of leaves co-converted with nYFP-VvMYC2 and cYFP-VvWDR1 and nYFP-VvMYBPA1 and cYFP-VvWDR1 using the BiFC in vivo (Figure 4B). However, fluorescence signals were not observed in the nYFP-VvMYBPA1 and cYFP-VvMYC2 group (Figure 4B).

Two results showed that VvMYBPA1 interacted with VvWDR1 but not with VvMYC2, and VvMYC2 interacted with VvWDR1. It was speculated that VvMYBPA1 promoted the amounts of total flavan-3-ols by forming a MBW trimer with VvMYC2 through VvWDR1 in grape.

### 2.5. VvMYBPA1 Could Be Directly Combined with the Promoters of VvLAR and VvANR

To explore the relationship between VvMYBPA1 and the key enzyme genes of flavan-3-ols biosynthesis, *VvLAR1* and *VvANR*, we conducted a Y1H experiment. The results indicated that the mutant group and empty vector could not grow on SD/Leu+ AbA^75 ng^ plate, but the test groups pGADT7-VvMYBPA1/pAbAi-pMBS(ANR), pGADT7-VvMYBPA1/pAbAi-pMBS(LAR1), grew on the SD/Leu+ AbA^75 ng^ plate using Y1H test (Figure 4C). The results indicated that VvMYBPA1 combined directly with the promoters of *VvLAR1* and *VvANR*.

## 3. Discussion

Previous studies found that UV-C could affect the accumulation of grape secondary metabolites, especially phenylpropanoid pathway. The significantly differential transcription factors *VvMYBPA1*, *VvMYB4*, *VvMYB14*, *WRKY57-like*, and *WRKY65* were screened through transcriptome analysis of grape leaves before and after UV-C treatment [10]. In this paper, we also found the increased expression of *VvMYBPA1* by UV-C treatment. The previous research of our lab also showed that UV-C would lead to the increase of LAR protein and LAR enzyme activity in juvenile fruit of grape, and finally increased the amounts of grape flavane-3-ols [23]. In this study, UV-C significantly improved the amounts of flavan-3-ol monomers in the juvenile grape fruit, and also promoted the large expression of *VvMYBPA1* at the young grape fruit (Figure 1). Therefore, we speculated that UV-C treatment improved the levels of flavan-3-ol monomers through increasing the expression of *VvMYBPA1* in the juvenile grape fruit.

The R2R3-MYB type transcription factor *VvMYBPA1* was not the most similar to AtTT2, a transcription factor related to PAs biosynthesis in *Arabidopsis*, which indicated that there was a certain difference between VvMYBPA1 and AtTT2. *VvMYBPA1* was mainly expressed in grape young fruit and leaves. No homologous gene of *VvMYBPA1* was found in *Arabidopsis*, but it could complement the phenotype of the tt2 mutant [16]. Passeri et al. found that the heterologous overexpression of *VvMYBPA1* into tobacco improved the content of PAs, and its related genes *NtANS* and *NtANR* were also largely expressed [26]. In this study, the accumulation of PAs and flavane-3-ol monomers increased the (the color of DAMSA staining was darker) instantaneous overexpression of *VvMYBPA1* as contrasted with the empty vector in the grape leaves (Figure 3A,B). We also found that the activities of ANR and LAR enzymes improved significantly (Figure 3D,E), and the expressed levels of *VvLAR1*, *VvLAR2*, and *VvANR* genes were higher in the transient overexpression of *VvMYBPA1* than the empty vector in the grape leaves (Figure 3C). Therefore, we found that *VvMYBPA1* influenced the transcription levels of flavan-3-ols biosynthesis-related structural genes (*VvLAR1*, *VvLAR2*, and *VvANR*) and their enzyme activities, which ultimately regulated the biosynthesis of flavan-3-ol monomers in grapes.

R2R3-MYB transcription factors, which regulated PAs and anthocyanin biosynthesis, generally interacted with bHLH and WDR to form a trimer to control the expression of their related structural genes [18]. Previous studies had proved that trimer regulated PAs and anthocyanin in various species. AtTT2 regulated PAs biosynthesis through a trimer with AtTT8 and AtTTG1 in *Arabidopsis* [27]. AcMYBF110, which formed a trimer with AcbHLH1 and AcWDR1, regulated the biosynthesis of anthocyanin in kiwifruit [28]. NtMYB330 regulated the biosynthesis of PAs through forming a trimer with NtAn1b and NtAn11-1 in tobacco [29]. PtoMYB134 and PtoMYB115 regulated the biosynthesis of PAs interacted with PtobHLH131 in poplar [13]. Research had indicated that WD40 was an alignment point for MYB and bHLH, enabling MYB-WD40-bHLH to establish trimeric complexes to affect the contents of anthocyanins [30]. *DkMYB4*, a *VvMYBPA1*-like gene, could form a MBW complex with *DkMYC1* and *DkWDR1*, and finally promoted the accumulation of PAs by regulating the expression of *DkANR* gene in persimmon fruit [19]. In this paper, the results indicated that VvMYBPA1 interacted with VvWDR1, but did not interact with VvMYC2, and VvMYC2 interacted with VvWDR1 using Y2H and BiFC. In conclusion, we speculated that VvWDR1 also provided a docking platform for VvMYBPA1 and VvMYC2 to form a trimeric complex to affect the content of PAs.

The structural genes of PAs biosynthesis directly encoded biological synthetases, namely LAR and ANR, and catalyzed the production of (+)-catechin and (−)-epicatechin, respectively. In general, MYB transcription factors enhanced the expression of structural genes by binding to related regulatory elements (MBS cis-element) in the promoter regions of downstream genes. There were some reports that indicated that NtMYB330 [29], GsMYB60 [14], DkMYB2/4 [19], MdMYB9/11 [12], and PtrBBX23 [31], etc., directly activated the expression of *ANR* or *LAR* in various species. *VvMYBPAR* also independently activated PAs synthesis-related genes *VvCHS3*, *VvF3’5’H*, and *VvLAR2* in grape [32]. However, several MYB transcription factors, which activated the promoters of structural genes, had to interact with bHLH and WD40 protein. In this paper, the results showed that VvMYBPA1 directly combined the MBS cis-elements on the promoters of *VvLAR1* and *VvANR* to regulate its expression and ultimately regulated the accumulation of PAs.

## 4. Materials and Methods

### 4.1. Plant Materials

An eight-years old ‘Zaoheibao’ grape (*Vitis vinifera* L. cv. ‘Zaoheibao’) with strong growth and no pests and diseases was cultivated in hedgerows, with a spacing of 1.5 m × 2.5 m and regular management. Nicotiana benthamiana was grown in a light incubator with a temperature of 25 °C, 16 h light/8 h dark, and humidity of 60–70%.

### 4.2. Grape Fruits at Different Stages Treated with UV-C

UV-C exposure test was slightly modified by consulting to the method of Niu et al. [9]. There were two treatments: no ultraviolet radiation (control) and 5 min each time. During the irradiation treatment, we set up a UV lamp group 1 m away from the plant to directly irradiate the plant. The UV lamp group was composed of two 40 W UV lamps (254 nm, ZSZ quartz UV germicidal lamp, Tianjin Guangze Special Light Source Co., Ltd., Tianjin, China), with a total radiation intensity of 95 μW·cm^−2^ (measured by ZQJ-ultraviolet radiation illuminometer). UV-C irradiation was performed every 10 days between 15 and 105 days after flowering. 30 panicles were gathered at the lower, middle, and upper parts of the panicle distribution area from five randomly selected plants. The fruit grains with mechanical injury, pests and diseases, and abnormal development were removed and stored at −80 °C for standby.

### 4.3. Instantaneous Overexpression in Grape Leaves

The RNA of ‘Zaohaibao’ grape fruits was collected by the modified CTAB method [25] and reverse-transcribed to cDNA. The total length of *VvMYBPA1* was amplified using primers containing cleavage sites and connected to the pMD19-T, incubated at 16 °C for 1 h, obtained a pMD-*VvMYBPA1* vector, and was converted into competent cells of E. coli Trans 5α. The pMD-*VvMYBPA1* vector and pCAMBIA1300 empty vector conducted a double enzyme digestion using *Hind* III and *Sal* I, then connected two enzyme cleavage products using T4 DNA ligase (Appendix A), constructed 35S::*VvMYBPA1* vector, and converted it into competent cells of Agrobacterium tumefaciens GV3101. Green grape leaves were infected with Agrobacterium tumefaciens containing empty and 35S::*VvMYBPA1* vectors for 10 min, respectively [24]. After infection, put them into a light incubator for culture. The culture temperature was 25 °C, the photoperiod was 16 h light/8 h dark, and the humidity was 60–70%. Samples were collected after 2–3 days and were stored at −80 °C for standby.

### 4.4. Fluorescence Quantitative Analysis

The modified CTAB method [25] was used to collect the RNA of UV-C treatment and control ‘Zaohaibao’ grape fruits at different stages, the transient overexpression of *VvMYBPA1*, and empty grape leaves reverse-transcribed to cDNA. The *VvMYBPA1* related to flavane-3-ol biosynthesis was analyzed by real-time fluorescent quantitative PCR using cDNA of UV-C treated and control ‘Zaohaibao’ grape fruit at different stages as template. Using transient overexpression of *VvMYBPA1* and empty leaf cDNA as templates, real-time fluorescent quantitative PCR analysis was conducted for *VvLAR1*, *VvLAR2*, *VvANR*, and *VvMYBPA1* genes. Grape *VvActin* was applied to internal reference (Appendix A). Two Realtime PCR Super mix (SYBR green, with anti Taq) kit instructions, pre-denatured at 95 °C for 1 min; denatured at 95 °C for 15 s, renaturation at 60 °C for 15 s, 40 cycles, fluorescence collection at 72 °C for 30 s, then calculated the relatively expression of related genes using the 2^−ΔΔCt^ method [25].

### 4.5. Subcellular Localization

The *VvMYBPA1* CDs without a terminator was amplified using the cNDA of young fruit of ‘Zaohaibao’ grape as the template, and the digestion site sequences were added in the primers (Appendix A), finally obtaining 35S::*VvMYBPA1*-GFP fusion vector. The *VvMYBPA1* amplified product was connected to pMD-19T (Japan TaKaRa Bio Company, Osaka, Japan) and was transferred into *E. coli* Trans 5α in the competent state (Beijing Quanshijin Biotechnology Co., Ltd., Beijing, China), and the plasmid was extracted with the plasmid small extraction kit (Tiangen Biochemical Technology (Beijing) Co., Ltd., Beijing, China) and sequencing. The plasmid and pCAMBIA1300 empty vector was digested with restriction endonuclease (Japan TaKaRa Bio Co., Ltd., Osaka, Japan), and connected with T4 ligase (Japan TaKaRa Bio Co., Ltd., Osaka, Japan), obtaining the 35S::*VvMYBPA1*-GFP plant expression vector. The fusion expression vector 35S::*VvMYBPA1*-GFP was transformed into the competent state of Agrobacterium tumefaciens GV3101 (Shanxi Shuoke Biotechnology Co., Ltd., Xi’an, China), and infected four weeks old tobacco leaves. Fluorescence signals were detected using laser confocal microscope (Leica TCS SP8, Leica, Wetzlar, Germany) after two/three days.

### 4.6. DMACA Staining

Grape leaves were infected by vacuuming according to the method of Li et al. and were slightly modified [33]. The *VvMYBPA1* and empty overexpressed leaves were immersed in the solution of glacial acetic acid–ethanol (1:3, *v*/*v*) for 24 h. Following that, the leaves were bleached using 75% ethanol for 12 h and were cleaned by double distilled water, then stained in the 0.6% DMACA for 2 min. They were finally examined and photos were for records. The depth of the staining color showed the content of PAs.

### 4.7. Content Determination of (−)-Epicatechin and (+)-Catechin

This was slightly modified with Wen’s method [2]. We accurately weighed 0.3 g of grape leaves ground with liquid nitrogen into a 10 mL centrifuge tube, added 70% methanol 3 mL, extracted them with ultrasound for 25 min, digested them for 12 h and centrifuged at 10,000 rpm for 10 min, and then obtained supernatant. The filtered supernatant carried out constant temperature rotary evaporation. 2 mL ethyl acetate and 1 mL water were added, the upper extraction phase was collected, drawn with 1 mL pure methanol, then filtered via 0.22 μM organic microporous membrane, and finally detected by high performance liquid chromatography (HPLC).

### 4.8. LAR and ANR Enzyme Activity Determination

The activities of LAR and ANR were determined as referred to by the method of Liang [25]. The protein content was determined by the Brandford method, and bovine serum protein was used as the standard curve. The expression method of LAR enzyme activity: the content of (+)-catechin produced by catalytic substrate within 1 h per milligram of protein was one LAR enzyme unit, U = mg CAT·mg^−1^·prot^−1^·h^−1^. ANR enzyme activity: the content of (−)-epicatechin, which was the product of catalytic substrate, within 20 min per milligram of protein, was one ANR enzyme unit, U = mg EC·mg^−1^·prot^−1^·min^−1^.

### 4.9. Yeast One Hybrid

The MBS cis-elements (MYB-binding site) of the *VvLAR1* and *VvANR* promoters were predicted based on the online website PlantCARE. The fragments, including MBS cis-element and mutation vectors, were infixed into the pAbAi vector and obtained the vectors pAbAi-p*VvANR*, pAbAi-p*VvLAR1*, and the mutation carriers pAbAi-pMBS(*anr*) and pAbAi-pMBS(*lar1*). The pGADT7 empty vector co-transformed into Y1Hgold competent cells with pAbAi(lar1), pAbAi(anr), pAbAi-ANR, and pAbAi-LAR1, respectively. They grew on the SD/-Leu medium with five different concentrations of Aureobasidin A (AbA), which contained AbA^0 ng/mL^, AbA^50 ng/mL^, AbA^75 ng/mL^, AbA^100 ng/mL^, and AbA^200 ng/mL^. The selected suitable concentration of AbA was 75 ng/mL (Appendix A). Total length of *VvMYBPA1* was inserted into pGADT7, obtained the pGADT7-VvMYBPA1 vector. Finally, pGADT7-VvMYBPA1/pAbAi-pMBS (LAR1), pGADT7-VvMYBPA1/pAbAi-pMBS(ANR), and negative group pGADT7-VvMYBPA1/pAbAi-pMBS(*lar1*), pGADT7-VvMYBPA1/pAbAi-pMBS(*anr*) were co-converted into Y1Hgold competent cells, and cultivated on selected medium SD/-Leu/AbA^75 ng/mL^. Following that, we observed the growth of plaque and took photos.

### 4.10. Yeast Two Hybrid

The total full-length of *VvMYBPA1*, *VvMYC2*, and *VvWDR1* were amplified and infixed into pGBKT7 and pGADT7 carriers and four plasmids, AD-MYBPA1, AD-MYC2, BD-WDR1, and BD-MYC2, were obtained. Three plasmid combinations (AD-MYBPA1/BD-WDR1, AD-MYBPA1/BD-MYC2, and AD-MYC2/BD-WDR1) were co-converted into AH109 receptive cells and cultivated on SD medium defect Trp-Leu for selection. The positive plaque grew on four deficient medium SD/- Trp-Leu-His-Ade, and blue on SD/- Trp-Leu-His-Ade-X- α- Gal medium. We observed and took photos to record the colony growth. The group of pGBKT7-53 and pGADT7-T was used as a positive group, and the group of pGBKT7-Lam and pGADT7-T was used as a negative group.

### 4.11. Bimolecular Fluorescence Complementation

The terminator-free *VvMYBPA1*, *VvMYC2*, and *VvWDR1* CDs sequences were amplified and connected to pCAMBIA1300-35S-YFPn and pCAMBIA1300-35S-YFPc carries, and we obtained four fusion vectors—nYFP-MYBPA1, cYFP-MYC2, cYFP-WDR1, and nYFP-MYC2. The constructed carriers were introduced into GV3101. The experimental groups were cYFP-WDR1 and nYFP-MYBPA1; cYFP-WDR1 and nYFP-MYC2; cYFP-MYC2 and nYFP-MYBPA1; the negative group cYFP and nYFP were injected into the 4-weeks old tobacco leaves, respectively. We observed and took photos after 2–3 days using laser confocal microscope (Leica TCS SP8, Leica, Wetzlar, Germany). DAPI indicated the nuclear of tobacco leaf cells.

### 4.12. Data Processing

The software of GraphPad Prism 9.0 was applied to data analysis and chart-making. MEGA 5.0 was used to construct phylogenetic tree by Neighbor-Joining method, 1000 bootstrapping. The software used for significant difference analysis and image arrangement were SAS8.0 and Photoshop CS6, respectively.

## 5. Conclusions

In this study, we found that UV-C treatment promoted the massive expression of *VvMYBPA1* and the amounts of flavane-3-ol monomers in the young grape fruit. The significantly differential transcription factors *VvMYBPA1* were also screened out through transcriptome analysis of grape leaves before and after UV-C treatment [10]. It was speculated that UV-C participated in the accumulation of total flavane-3-ols at the young fruit stage through positively regulating the expression of *VvMYBPA1*. Our results demonstrated that a trimeric complex was formed by VvMYBPA1 and VvMYC2 through VvWDR1, then promoted the expression level of structural genes *VvLAR1* and *VvANR*, then elevated enzyme activities of LAR and ANR, improved the accumulation of flavane-3-ol monomers, and finally increased the content of PAs (Figure 5). However, the relationship between UV-C and *VvMYBPA1* is currently not very clear and further research is needed.

## Figures and Tables

**Figure 1 plants-12-01691-f001:**
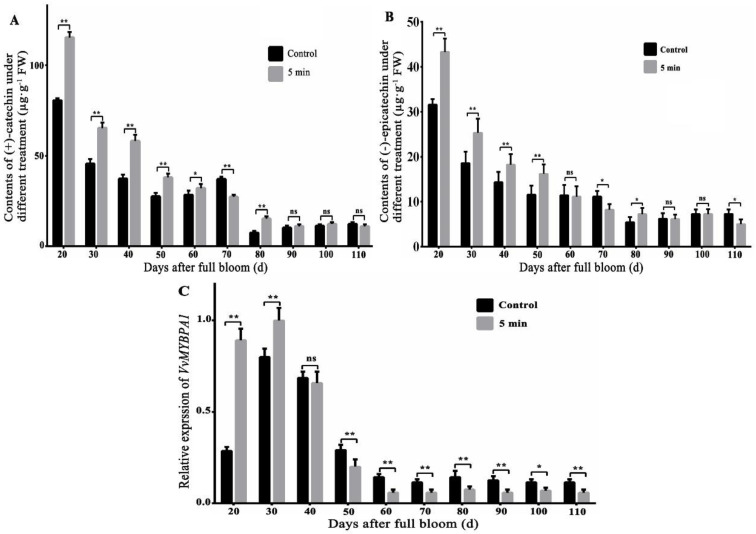
The contents of grape flavane-3-ol monomers and the expression of *VvMYBPA1* of 5-min UV-C treatment and control at different stages of fruit development. (**A**) The contents of (+)-catechin of UV-C treatment and control at different stages of fruit development, (**B**) the contents of (−)-epicatechin, (**C**) the expression of *VvMYBPA1* of UV-C treatment and control. Note: ** indicated extremely significant differences at the level of *p* < 0.01; * indicated significant differences at the level of *p* < 0.05, while ns indicated no significant difference.

**Figure 2 plants-12-01691-f002:**
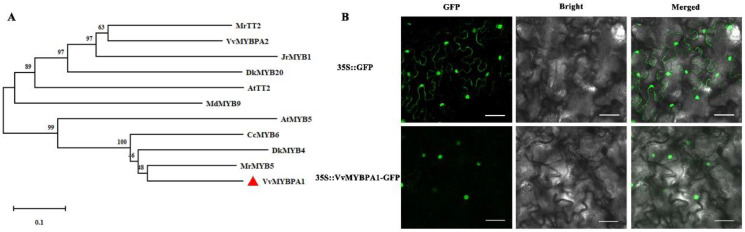
VvMYBPA1 characteristic analysis. (**A**) Phylogenetic analysis of VvMYBPA1, (**B**) Subcellular localization of VvMYBPA1.

**Figure 3 plants-12-01691-f003:**
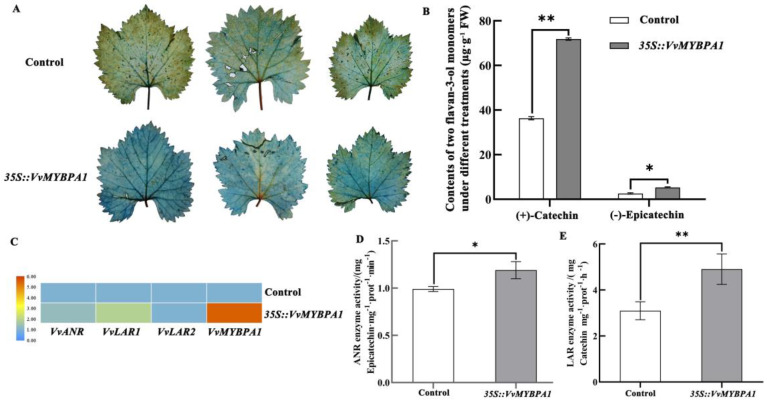
The differences of the accumulation of grape flavane-3-ol monomers, the expression of its related genes, and the enzyme activity of related enzymes in the grape leaves of instantaneous overexpression of *VvMYBPA1* and the empty vector. (**A**) DAMAS staining of the grape leaves of transient overexpression of *VvMYBPA1* and the controls; (**B**) The contents of (+)-catechin and (−)-epicatechin in the grape leaves of transient overexpression of *VvMYBPA1* and the controls; (**C**) The expression of *VvLAR1*, *VvLAR2*, *VvANR*, and *VvMYBPA1* in the grape leaves of transient overexpression of *VvMYBPA1* and the controls; (**D**,**E**) LAR and ANR enzyme activity in the grape leaves of transient overexpression of *VvMYBPA1* and the controls. Note: ** indicated extremely significant differences at the level of *p* < 0.01; * indicated significant differences at the level of *p* < 0.05.

**Figure 4 plants-12-01691-f004:**
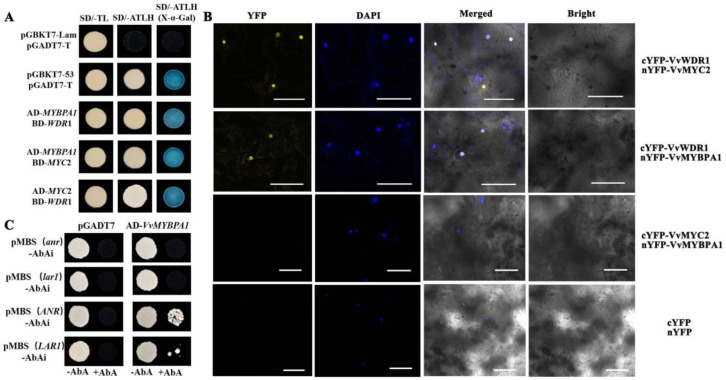
Interaction between VvMYBPA1 and flavane-3-ols-related genes. (**A**) Yeast two hybrid between VvMYBPA1, VvMYC2, and VvWDR1; (**B**) BiFC between VvMYBPA1, VvMYC2, and VvWDR1; (**C**) Y1H between VvMYBPA1 and the promoters of *VvLAR* and *VvANR*.

**Figure 5 plants-12-01691-f005:**
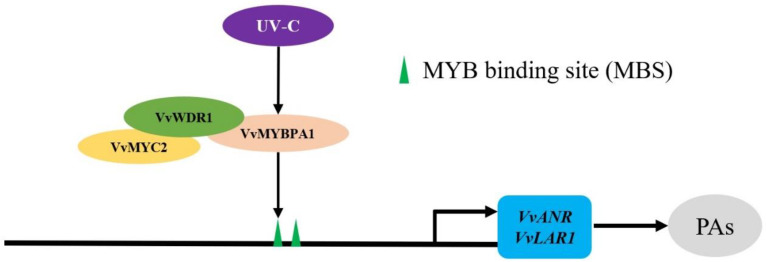
Prediction of the biosynthesis pattern of grape PAs regulated by MBW complex with UV-C.

## Data Availability

The data presented in this study are available on request from the corresponding author.

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
