# Peer review of "UV-C Promotes the Accumulation of Flavane-3-ols in Juvenile Fruit of Grape through Positive Regulating VvMYBPA1"

_plants, 2023, doi:10.3390/plants12081691_

Round 1

Reviewer 1 Report

It is a quite good research adding some new details concerning PAs biosynthesis and its regulation.

Author Response

Thank you very much for your suggestion. The English language and style of the article has been modified.

Reviewer 2 Report

Suggestions,

1.     How VvMYBPA1 was amplified and inserted into the pCAMBIA1300? Which cloning method? Needs to elaborate.

2.     2-△△Ct method. Reference for this? 

3.     Please provide raw data for RT-PCR study as a supplementary file.

4.     List all the primers in a supplementary file.

5.     Modified CTAB method for RNA extraction? Reference for this?

6.     How quality of RNA tested?

7.     The manuscript needs revision for language and grammar.

8.     Future prospectus is not mentioned in conclusion.

9.     Scientific names are always italicized. Please check.

10.   Which restriction enzymes used?

Author Response

Dear reviewer:

Thank you very much for your suggestion. Your suggestion were very relevant and had been revised in the corresponding position of the manuscript. The attached Word file contains responses to all your questions.

Best regards

Jinjun Liang

Reviewer 3 Report

The authors present an accurate, well-developed investigation of the molecular components that bring PA production in grapes. The experiments are well-designed to demonstrate the causative relationship between TFs, Enzymes, and metabolites. 

I have a few minor comments; for the rest, I recommend the publication of the manuscript in its current state.

 1. Section 2.12 - Specify your methods to generate the phylogeny tree. In particular, what distance metrics, method (is this NJ or another?) and bootstrapping did you apply? What did you use in MEGA?

2. Figure 3C- Please add the y-axis and legend on the left side. Right now its overlapping with the 3D and its a bit confusing

3. Same for Figure 4B - could you move the y-axis labels on the right side?

4. Pag 7 - "In order to find out the reason of VvMYBPA1 led to the content of total flavane-3-ols in grape." this is a partial sentence; correct it to a full sentence. 

Author Response

Dear reviewer:

Thank you very much for your suggestions. Your suggestions were very relevant and had been revised in the corresponding position of the manuscript. The attached Word file contains responses to all your questions.

Best regards

Jinjun Liang

Round 2

Reviewer 2 Report

Authors has addressed comments raised by.